# Causal Association of Free Triiodothyronine Level with Ischemic Stroke Outcome: A Mendelian Randomization Study

**DOI:** 10.3390/life15081303

**Published:** 2025-08-16

**Authors:** Dmitry A. Filimonov, Vitaly V. Morozov, Roman V. Ishchenko, Alexander B. Eresko, Nadezhda N. Trubnikova, Margarita A. Belotserkovskaya, Maksim V. Solopov, Irina A. Kisilenko, Inna N. Nosova, Dmitry A. Kudlay

**Affiliations:** 1Federal State Budgetary Institution “V.K. Gusak Institute of Emergency and Reconstructive Surgery” of the Ministry of Health of the Russian Federation, Donetsk 283045, Russia; info@iursdn.ru (R.V.I.); n.trubnikova@iursdn.ru (N.N.T.); m.belotserkovskaya@iursdn.ru (M.A.B.); m.solopov@iursdn.ru (M.V.S.); i.kisilenko@iursdn.ru (I.A.K.); i.nosova@iursdn.ru (I.N.N.); 2The Institute of Chemical Biology and Fundamental Medicine of the Siberian Branch of the Russian Academy of Science, Novosibirsk 630090, Russia; morozov_vv@niboch.nsc.ru; 3Joint Institute for Nuclear Research, Dubna 141980, Russia; a_eresko77@jinr.ru; 4Department of Pharmacology, Federal State Autonomous Educational Institution of Higher Education I.M. Sechenov First Moscow State Medical University of the Ministry of Healthcare of the Russian Federation (Sechenovskiy University), Moscow 119991, Russia; admission@staff.sechenov.ru

**Keywords:** ischemic stroke, thyroid hormones, Mendelian randomization, genetic variants, triiodothyronine, neuroprotection

## Abstract

The global burden of ischemic stroke requires a search for new factors that affect its risk and outcomes. Recent studies have shown that triiodothyronine could potentiate favorable stroke outcomes, but the reason for this is still unclear. To clarify the effects of the free triiodothyronine (fT3) level on stroke outcomes, we adopted a Mendelian randomization (MR) approach to evaluate their causal relationship. The genetic variants associated with the free triiodothyronine (fT3) level were obtained from the ThyroidOmics Consortium. Thirteen single-nucleotide polymorphisms, genetically predicting the fT3 level with a significance of *p* < 10^−7^, were adopted for MR analysis. Summary-level data for ischemic stroke outcomes (reported as a continuous variable, namely the modified Rankin score 3 months after stroke) was obtained from the GISCOME network. MR analyses were performed using the TwoSampleMR framework. The inverse-variance weighting method of MR analysis showed that a genetically predicted increase in fT3 level is associated with a reduction in ordinal Modified Rankin Scale scores (OR = 0.581, 95% CI 0.37–0.92, *p* = 0.0183). This study showed that higher fT3 levels could be causally associated with more favorable ischemic stroke outcomes and provides more evidence for the possibility of using thyroid hormone (TH) analogs to improve stroke outcomes.

## 1. Introduction

Ischemic stroke is a major medical and social problem. Every year, 12.2 million strokes occur worldwide [1], remaining one of the leading causes of death and disability. More than half of stroke survivors experience a variety of outcomes that impair quality of life, reduce life expectancy, and limit their somatosensory and cognitive functions, movement, and speech [2,3].

Reperfusion therapy significantly improves both the rate of vascular recanalization and functional outcomes in patients with acute ischemic stroke. However, a substantial number of patients are either unsuitable for recanalization therapy or experience limited recovery. The gold standard for treating patients with acute ischemic stroke is intravenous thrombolysis with a tissue plasminogen activator. However, this is only effective within 4.5 h of the onset of symptoms. Intravenous thrombolysis is less effective for larger arterial occlusions, achieving recanalization in only 20% of such cases. Mechanical thrombectomy, indicated for the occlusion of a large artery in the anterior circulation, is effective within 24 h; however, comparative studies which have examined its effect on long-term stroke outcomes have shown conflicting results, since less than 50% of patients achieve functional independence [4,5,6]. Reperfusion restores cerebral blood flow to the hypoperfused brain region surrounding the infarcted core, sparing penumbra cells. The penumbra shrinks over time; this is seen in 90–100% of stroke patients in the first 3 h after stroke onset and in about 33% of patients 18 h after stroke onset [7]. For reperfusion therapy to be successful, additional neuroprotective measures must delay cell death until recanalization is achieved. Neuroprotective strategies may potentially address some of the limitations of reperfusion therapy and improve its effectiveness, and there is still a need for effective neuroprotective methods to improve the outcomes of acute ischemic stroke. Early intervention with neuroprotective therapies has the potential to limit damage and enhance functional recovery [8].

In the field of pharmacological neuroprotection, over the past 15 years, there has been a trend of using various endogenous substances and their synthetic analogs [9]. The brain has high sensitivity to ischemia; therefore, various endogenous mechanisms for its protection have been formed evolutionarily. When ischemia develops, these mechanisms should be triggered naturally, but their activity can likely be increased through the exogenous administration of appropriate substances.

Thyroid hormones (THs) could be promising endogenous neuroprotectors. They are essential for the normal functioning of the central nervous system, and it has been suggested that THs may regulate brain-derived neurotrophic factor (BDNF) gene expression and synaptic plasticity, as well as influencing neuroinflammation and apoptosis processes [10].

THs levels affect the growth and maturation processes of neurons—in particular, the processes of migration, cell differentiation, dendritic process expansion, synaptogenesis, and nerve cell myelination [11,12]. It has been indicated that during brain reorganization after a stroke, mechanisms of brain development may be reactivated, and the genes involved in these processes are regulated by THs [13].

A number of clinical studies have demonstrated certain associations between low triiodothyronine (T3) syndrome and ischemic stroke outcome. In 2017, the results of two meta-analyses including 3936 and 5218 patients were published [14,15]. Both studies demonstrated an association between low free triiodothyronine (fT3) levels and adverse functional outcomes of ischemic stroke, while Dhital et al. reported that hypothyroidism, defined as elevated thyroid-stimulating hormone (TSH) levels, could be associated with a more favorable functional outcome. However, a causal relationship between T3 levels and stroke functional outcomes has not yet been established. Evidence of such a causal relationship could be an important basis for initiating pilot studies on the efficacy of T3 analogs in improving stroke outcomes. Obtaining such evidence may require large sample sizes, and the results may be affected by various cofactors and reverse causation. To overcome these possible limitations, we used a novel method for assessing risk factor influence on disease outcome—Mendelian randomization (MR) [16,17].

The last decade has seen an increase in scientific research using MR, which evaluates the effect of an exposure on an outcome using genetic variants associated with the exposure as instruments [18]. MR provides a less biased approach to studying the effects of treatment on outcomes than traditional observational epidemiological methods [19]. The most popular method of MR analysis is the two-sample approach, based on the use of publicly available large-sample genome-wide association study (GWAS) databases [20,21]. Instrumental variables (IVs) or single-nucleotide polymorphisms (SNPs) used in MR must meet three conditions (Figure 1): a strong relationship with exposure (risk factor), influence on the result only through their impact, and the absence of factors affecting both the SNP and the result.

To analyze the effect of a risk factor (trait 1) on an outcome (trait 2), a linear regression of the effects of SNPs on trait 1 versus the effects of SNPs on trait 2 is used, with the regression coefficient as an estimate of the causal impact of trait 1 on the resulting trait 2. Depending on the presence or absence of SNP pleiotropic effects on the traits under study, an appropriate regression model should be selected: the inverse-variance-weighted linear regression model works only in the absence of pleiotropy [22,23], and various methods have been developed that take into account genetic pleiotropy. These help to determine whether the result depends on the assumption that all variants do not have pleiotropic effects. One such approach, known as the median score, can give reliable results if at least half of the genetic variants have no pleiotropic effects. The second method, known as Egger’s regression, allows all variants to have pleiotropic effects, provided that the size of these pleiotropic effects is independent of the size of the genetic variants’ effects on the risk factor of interest.

Thus, MR studies can provide reliable evidence for the influence of risk factors on disease and can overcome some of the limitations of traditional observational epidemiology.

To our knowledge, this is the first MR study aiming to determine relationship between fT3 level and stroke functional outcome. The purpose of this study is to determine the possible causal effect of fT3 levels on the severity of ischemic stroke using MR analysis.

## 2. Materials and Methods

### 2.1. Selection of Genetic Variants Associated with Exposure

The flowchart of our MR study is displayed in Figure 2.

We used data on genetic variants associated with T3 level from a GWAS analysis by the ThyroidOmics Consortium, and genome-wide significant associations for fT3 levels were taken from the Sterenborg R. et al.’s dataset (https://genepi.med.uni-greifswald.de/thyroidomics/datasets/ (accessed on 11 August 2025)) [24]. A total of 271,040 euthyroid individuals with European ancestry were included in their study investigating thyroid function, for which they collected data on the reference range of TSH, free thyroxine (fT4), free and total triiodothyronine (fT3 and T3), T3/fT4 ratio, etc. In total, 59061 euthyroid individuals were included in their study of fT3 levels.

Since we used only GWAS summary data, ethical committee approval was not required for this study.

The most commonly accepted threshold in GWASs is *p* < 5 × 10^−8^. We had to use a less strict SNP significance approach (*p* < 5 × 10^−7^) because of the small number of available SNPs associated with fT3 levels. The 14 SNPs that meet this requirement are presented in Table 1. We utilized the GWAS catalog online service (https://www.ebi.ac.uk/gwas/ (accessed on 11 August 2025)) to assess whether the selected SNPs were associated with other traits at genome-wide significance levels, potentially violating the second and third key assumptions, and did not find any such violations. The F-statistics of the selected IVs were all < 10, the threshold of weak instruments, indicating strong IVs for the MR [25,26].

### 2.2. Data Sources for Stroke Outcome

Single-nucleotide variations associated with stroke severity were retrieved from the GISCOME database (Genetics of Ischemic Stroke Functional Outcome network) containing data from 12 GWASs conducted by the International Consortium of Stroke Genetics (International Stroke Genetics Consortium) and the Stroke Initiative Genetics Network of the National Institute of Neurological Diseases (USA) [27]. The GISCOME database includes data on the genotype and ischemic stroke outcome of 8831 patients; we used the outcome data, unadjusted for baseline stroke severity (https://kp4cd.org/node/391 (accessed on 11 August 2025)).

The GISCOME database used the Modified Rankin Scale (mRs) as a variable reflecting stroke severity. We utilized data on the associations of SNPs with ordinal Modified Rankin Scale (mRS) scores across the full spectrum. This approach enables the assessment of a factor’s influence on different degrees of functional outcome and, as noted by the authors of the original study, offers greater power.

### 2.3. Statistical Analysis

We performed two-sample MR analyses using the inverse-variance-weighted, MR-Egger, and weighted median methods to detect the results’ robustness [27,28]. We also performed a leave-one-out sensitivity analysis to assess whether the results were influenced by individual SNPs. Cochran’s Q statistic was used to detect heterogeneity in the inverse-variance-weighted and MR-Egger analyses.

The analyses were conducted using R version 4.4.3, employing TwoSampleMR (R package version 0.5.6) in the RStudio (version 2024.09.1) development environment. All the statistical tests were two-tailed.

### 2.4. Data Availability

All GWAS summary datasets analyzed during this study are publicly available. The ThyroidOmics Consortium only provides GWAS summary data for scientific purposes if the researcher does not attempt to re-identify individual participants included in the summary statistics. Materials from the article of Sterenborg et al. are licensed under a Creative Commons Attribution 4.0 International License. GISCOME datasets are available on The Common Metabolic Diseases Knowledge Portal, and all users are welcome to use results from analyses of these data labeled “Open access” to further their research without seeking explicit permission.

## 3. Results

Rs1588635 and rs150816132, which were associated with fT3 level, were absent in the GISCOME database, so we searched for proxy SNPs. For rs1588635, a proxy rs7028661 was found. Rs150816132 did not have available proxies, so this SNP was excluded from further study. After linkage disequilibrium clumping, we finally identified 13 SNPs as instrumental variables in our MR analyses, since the rest were absent in the GISCOME database without available proxies (Figure 3).

The genetically predicted increase in fT3 levels was statistically significantly negatively associated with risk of unfavorable ischemic stroke outcome when analyzed using the inverse-variance-weighted method (*p* = 0.018, b = −0.543) (Figure 4). The odds ratio for unfavorable ischemic stroke outcomes was OR = 0.581, with the 95% CI = 0.37–0.91. The results were similar in weighted-median-based sensitivity analysis (*p* = 0.074, b = −0.599).

To visualize the magnitude of the effect calculated using various Mendelian randomization analysis methods, a scatterplot was constructed (Figure 5). Each of the SNPs associated with fT3 levels is represented by a black dot, with error bars depicting the standard errors of its association with fT3 level (horizontal) and stroke severity (vertical), and effects presented as β. The slopes of the lines are the causal relationships calculated using different MR methods, providing comparisons between them.

The leave-one-out sensitivity analysis (Figure 6) indicated that no SNP altered the MR estimates, suggesting the stability and reliability of the forward MR results.

The MR–Egger test showed no evidence of horizontal pleiotropy for the effects of triiodothyronine levels on ischemic stroke outcome (*p* = 0.345).

A test for heterogeneity was also performed, which, under Mendelian randomization, is a statistical assessment of the compatibility of instrumental variable estimates based on individual genetic variants. Cochran’s Q statistic was used, with Q values greater than the degrees of freedom (number of instrumental variables minus 1) indicating heterogeneity and invalid instruments. For the inverse-variance-weighted linear regression model, a Cochrane Q statistic of 12.53, with a corresponding *p* value of 0.404, showed no strong evidence of heterogeneity.

## 4. Discussion

The strengths of this study include its two-sample MR design, large sample size, and use of multiple methods of analysis.

For most of the SNPs used as instrumental variables in the MR analyses, their associations with any traits and clinical significance have not been described. An exception is rs1169288—a missense variant of HNF1A (hepatocyte nuclear factor 1-alpha)—whose associations with low-density lipoprotein cholesterol, total cholesterol, and gamma glutamyl transferase levels were found in GWAS and SNP databases (EMBL-EBI, SNPedia). In turn, gamma-glutamyl transferase is involved in maintaining the physiological concentration of glutathione in cells and protects them from oxidative stress-related damage.

The results of our study confirm the data of a number of studies devoted to the prognostic significance of blood serum fT3 levels in patients with acute ischemic stroke [11,29,30]. In 2018, a meta-analysis of publications on this issue was carried out. The authors of the analysis concluded that a decrease in the blood serum fT3 level of patients with acute ischemic stroke is a reliable prognostic criterion that directly correlates with the likelihood of an unfavorable outcome [31]. Li et al. found that the association between low T3 levels and poor prognosis after acute stroke was more relevant for patients over 65 years old [32]. In the study by Zhang S., the T3/T4 ratio was an independent risk factor for all-cause mortality in stroke survivors [33].

There are several putative mechanisms of T3 influence on the course of ischemic stroke. First of all, T3 realizes most of the genomic effects of THs by directly entering the nucleus and interacting with nuclear receptors. Non-genomic mechanisms are activated by both T3 and T4, as well as other THs. These hormones can also affect transcriptional activity through a non-genomic mechanism of action, but their effect extends to a larger number of genes [34].

THs are suggested to influence neuronal regeneration. A single administration of T3 24 h after transient middle cerebral artery occlusion in rats increased the expression of brain-derived neurotrophic factor, nestin, and SRY-box transcription factor 2 [1], and the administration of T3 in the acute phase of ischemia in the experiment led to a decrease in the expression of aquaporin-4, a water channel protein that is involved in the formation of cerebral edema after ischemic stroke [1].

Reduced fT3 levels are associated with the dysregulation of a number of genes encoding mitochondrial proteins and proteins involved in mitochondrial function. Mitochondria are one of the major sources of reactive oxygen species (ROS), as well as the major sites of T3 accumulation in cells [35]. Physiological levels of ROS are necessary for normal cell function, but increased levels lead to cell damage and death, meaning that reliable antioxidant mechanisms are important for cells with high levels of ROS (such as brain cells), [35]. This dependence is nonlinear and organ-specific, but, in general, it is believed that hypothyroidism decreases ROS production, while hyperthyroidism leads to its increase, accompanied by a decrease in antioxidant activity [36,37].

In the study of the mechanisms by which THs can influence oxidant status, special attention is paid to T3’s activating effect on uncoupling protein 2 (UCP2) and uncoupling protein 3 (UCP3) gene expression [38,39]. Although these genes are orthologs of uncoupling protein 1 (UCP1), a thermogenin that enables energy-to-heat conversion in adipocytes, their function is not related to uncoupling. The T3-induced activation of UCP2 and UCP3 correlates with increased mitochondrial calcium uptake, which in turn affects ROS production [40]. Furthermore, THs are involved in the regulation of antioxidant proteins such as superoxide dismutase, catalase, glutathione peroxidase, glutathione-S-transferase, and cytochrome oxidase [40].

Additionally, T3 is required for the elimination of mitochondria damaged by oxidative stress due to its action on peroxisome receptor coactivator gamma-1 [41]. It is known that the experimental induction of hypothyroidism increases mitochondrial vacuolization, decreases the number of cristae, and changes the levels of proteins forming respiratory chain complexes, which leads to functional changes: a loss of transmembrane potential and reduction in electron transport chain capacity [42,43,44]. Dysfunctional mitochondria produce the most ROS and cause the release of cytochrome c (cyt c), leading to cell death, while T3 restores mitochondrial integrity, normalizes respiratory processes, and regulates mitochondrial transcript levels [45].

The effect of thyroid hormones on the synthesis of ion exchange channel proteins is considered to be proven. For example, T3 activates the expression of the Na/H exchanger, which removes excess protons from cells and maintains normal pH, as well as increasing the incorporation of Na/K-ATPases into the cell membrane, which is required to maintain low intracellular Na^+^ levels [46,47].

Another possible iodothyronine neuroprotective mechanism is the activation of basic fibroblast growth factor (bFGF) secretion, as the study of bFGF function in a transient global cerebral ischemia model revealed a significant neuroprotective effect [48]. In this case, the signal is transmitted not from nuclei but from membrane receptors through the extracellular signal-regulated kinase (ERK 2) chain [49].

In an experimental model of traumatic brain injury, T3 administration increased the expression of *Tet* family genes and decreased the expression of DNA methyltransferases (Dnmt) 3a and Dnmt3b, blocking hypoxia-induced DNA methylation which reduced neuronal damage and prevented apoptosis [50,51].

After passing through the acute phase of stroke, the brain is capable of partially recovering lost neurological functions. THs regulate several pathways involved in neural tissue repair and related to the processes of neuroplasticity, neurogenesis, angiogenesis, and glutamate toxicity. For example, T3 regulates the activity of the Nrgn gene, which encodes a calmodulin-binding protein that enhances synaptic plasticity. THs also regulate the expression of rilin, which participates in the migration of multipolar neurons in the embryonic period and regulates synaptic plasticity and neurogenesis in adulthood [52].

Despite the small number of studies concerning T3, they all point to its involvement in signaling pathways that contribute to the protection and repair of neurons through genomic or non-genomic mechanisms, and a large amount of data has now been accumulated on T3’s potential neuroprotective effects [48,53]. At the same time, clinical studies on the effectiveness of synthetic T3 analogs in improving stroke outcomes have not yet been conducted, despite the fact that some of these analogs, such as liothyronine, have an acceptable safety profile [54]. In addition to existing data, our study provides another reason to initiate such clinical trials.

### Limitations

This study has limitations. First of all, we had a limited amount of instrumental variables: we had to use a less strict SNP significance threshold (*p* < 10^−7^) because few SNPs related to fT3 level with traditional genome-wide significance *p* < 10^−8^ are available, and these slightly relaxed thresholds applied to our MR analysis may have introduced false positives. However, in our MR analysis, the *F*-statistics of the selected SNPs exceeded 10, indicating the robustness of the instrumental variables. Also, we did not have enough GWAS information to use the MR Steiger directionality test to definitely rule out reverse causality. Still, MR itself allows the exclusion of reverse causality with a very high degree of probability. This study also only included Europeans, so it may not be fully representative of other ethnicities. Furthermore, we could not completely exclude the possibility of bias due to pleiotropy, despite a range of sensitivity analyses.

## 5. Conclusions

This study showed that higher free triiodothyronine levels could be causally associated with more favorable ischemic stroke outcomes. Although the SNPs used in this study as instrumental variables do not explain the underlying mechanism of this association, it is likely related to genetic factors, and further clinical studies are advisable to validate these associations and assess their practical significance. In addition to recent clinical and experimental studies, this study provides more evidence for the possibility of using thyroid hormone analogs to improve ischemic stroke outcomes.

## Figures and Tables

**Figure 1 life-15-01303-f001:**
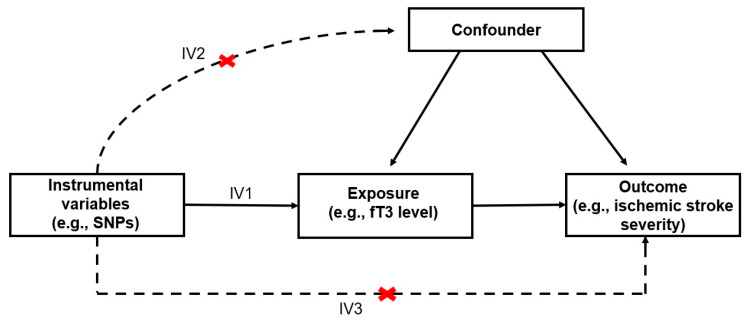
Instrumental variable assumptions for our Mendelian randomization study. IV1—association, IV2—no association, IV3—association only via exposure.

**Figure 2 life-15-01303-f002:**
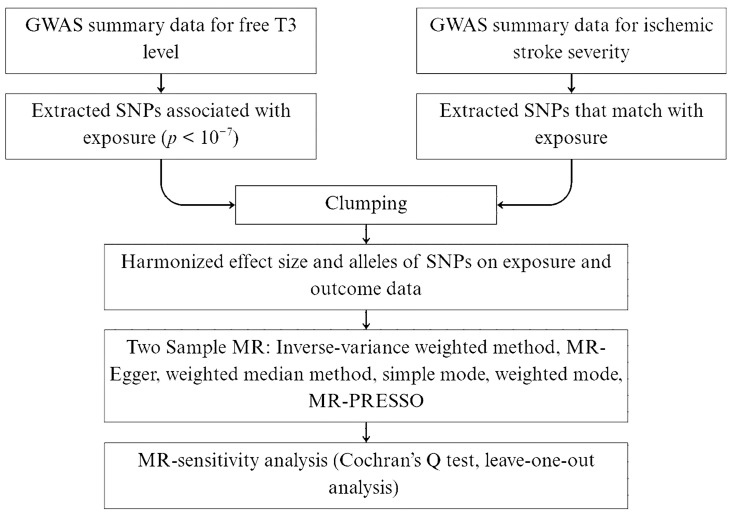
The flowchart of the Mendelian randomization study to determine the relationship between free triiodothyronine level and stroke functional outcome.

**Figure 3 life-15-01303-f003:**
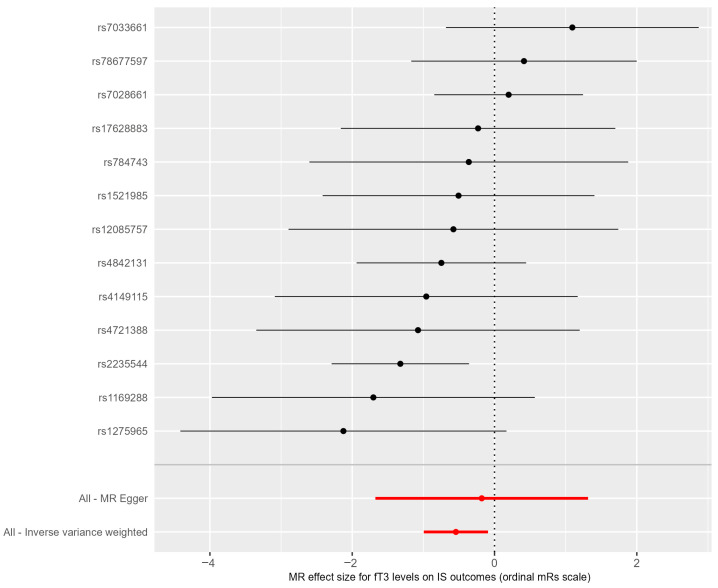
A forest plot of the MR effect size of free triiodothyronine levels on the risk of unfavorable ischemic stroke outcome. Black lines show causality calculated using each SNP separately, red lines show causality calculated using methods that use all SNPs.

**Figure 4 life-15-01303-f004:**
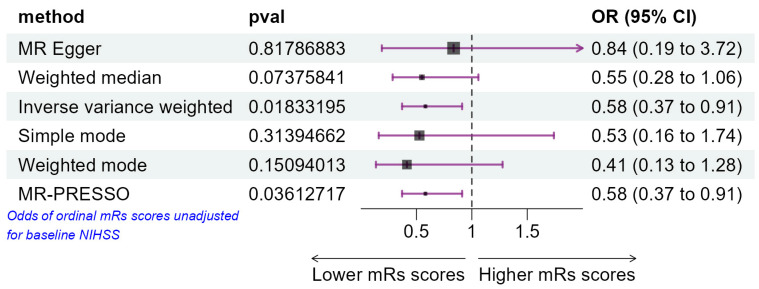
Odds ratio plot for fT3 levels from different MR tests. Note: pval—meta-analysis *p*-value using regression coefficients; OR—odds ratio. The odds ratios of fT3 levels on the risk of worse ischemic stroke outcome (higher mRs scores) are displayed as a black solid box. The 95% CIs are shown as horizontal violet lines.

**Figure 5 life-15-01303-f005:**
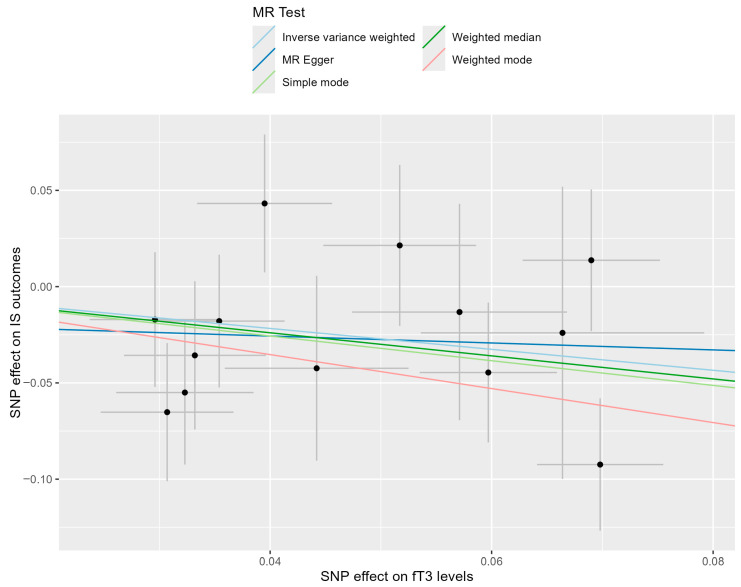
A scatterplot showing the relationship between the effects of SNPs on fT3 levels and the risk of unfavorable ischemic stroke outcome. Each black dot represents a valid instrumental SNP. The horizontal and vertical short lines through the dots represent the 95% CIs of the SNP effect on free triiodothyronine level and the risk of higher mRs scores (worse outcome), respectively. The slope of fitted lines represents the estimated causal effect of free triiodothyronine level on the risk of higher mRs scores (worse ischemic stroke outcome).

**Figure 6 life-15-01303-f006:**
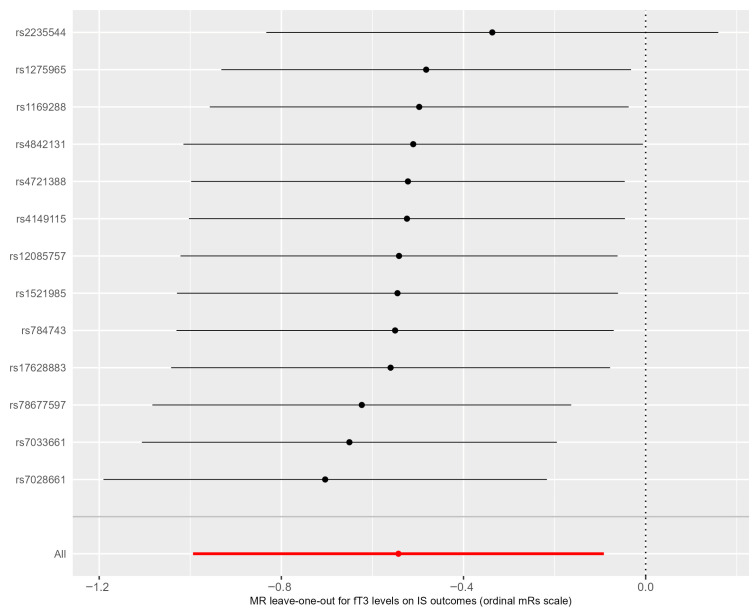
A leave-one-out plot for the MR sensitivity analysis. The dots represent the odds ratios after removing the corresponding SNP, and the lines represent the corresponding 95% confidence intervals. Black lines show causality calculated using each SNP separately, red line shows causality calculated using methods that use all SNPs.

**Table 1 life-15-01303-t001:** Single-nucleotide polymorphisms associated with the level of free triiodothyronine.

SNP	Chr	Gene	EA	OA	FREQ	BETA	SE	*p*-Value
rs1169288	12	HNF1A	A	C	0.6786	−0.0323	0.0062	2.205 × 10^−7^
rs4149115	12	SLCO1B3	A	G	0.1488	0.0442	0.0083	8.87 × 10^−8^
rs150816132	14		A	G	0.012	0.1563	0.029	6.832 × 10^−8^
rs12085757	1	YIPF1	T	C	0.3789	0.0296	0.0059	4.488 × 10^−7^
rs2235544	1	DIO1	A	C	0.5227	−0.0698	0.0057	1.865 × 10^−34^
rs1275965	2		T	C	0.3852	0.0307	0.006	3.022 × 10^−7^
rs784743	3		T	C	0.9455	0.0664	0.0128	2.006 × 10^−7^
rs17628883	4	AADAT	A	G	0.1027	−0.0571	0.0097	3.89 × 10^−9^
rs1521985	5		T	C	0.4854	−0.0354	0.0059	2.711 × 10^−9^
rs78677597	6	LOC105377911	A	C	0.7704	−0.0517	0.0069	8.487 × 10^−14^
rs4721388	7	DGKB	A	C	0.695	0.0332	0.0064	2.254 × 10^−7^
rs1588635	9	PTCSC2	A	C	0.3431	0.069	0.0062	1.435 × 10^−28^
rs7033661	9	ZNF462	A	G	0.6435	−0.0395	0.0061	1.375 × 10^−10^
rs4842131	9	LHX3	T	C	0.4353	−0.0597	0.0062	4.608 × 10^−22^

Note: SNP—single-nucleotide polymorphism; Chr—chromosome; EA—effect allele (coded allele); OA—non-effect allele (non-coded allele); FREQ—effect allele frequency; BETA—overall estimated effect size for effect allele; SE—overall standard error for effect size estimate; *p*-value—meta-analysis *p*-value using regression coefficients.

## Data Availability

The generated data presented in this study are available from the corresponding author on request.

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
