# Peer review of "Causal Association of Free Triiodothyronine Level with Ischemic Stroke Outcome: A Mendelian Randomization Study"

_life, 2025, doi:10.3390/life15081303_

Round 1
Reviewer 1 Report
Comments and Suggestions for Authors
Figure 2 should be inserted after the first mention (between lines 113 and 114).
Table 1 is not mentioned in the main text of the manuscript. The main data presented in this table should be described in the text.
Please explain the reasons for the discrepancies between the lists of single nucleotide polymorphisms (SNPs) given in Table 1 and Figures 3 and 6. I did not find rs7028661 in Table 1, although this SNP is presented in Figures 3 and 6. Table 1 also lists rs150816132 and s1588635 that not included in the figures. Please, explain if these specific SNP were removed.
Section 5 should be expanded to include the main conclusion on the SNPs, since elucidation of the relationship between SNPs and free T3 levels was one of the main objectives of the work.
Minor corrections:
Add abbreviations for ROS, SOX2, TH, etc. after the first mention.
Use the correct term "exchanger" instead of "ex-changer" (line 258).
Capitalize the institute name (line 11).
Author Response
Dear Reviewer,
We are writing to express our deepest gratitude for your thorough and insightful review of our manuscript, "Causal association of free triiodothyronine level with ischemic stroke outcome: a mendelian randomization study". Your expertise and constructive suggestions have been invaluable to our work.
We wish to emphasize that all your comments and recommendations were absolutely relevant and have significantly strengthened the quality of our paper.
In response to your comments, we have revised the manuscript and prepared a point-by-point response to each of your suggestions. The updated manuscript has been submitted via the journal’s editorial system. We hope the revisions meet your expectations and adequately address all the points you raised.
In accordance with these comments, we have made the following changes to the article.
- We have moved Figure 2 directly after the paragraph where it is mentioned.
- We have added a description of table 1: «14 SNPs that meet this requirement are presented in Table 1.» (lines 139-140).
- Rs1588635 and rs150816132, which were associated with fT3 level, were absent in the GISCOME database, so we searched for proxy SNPs. For rs1588635, a proxy rs7028661 was found. Rs150816132 does not have available proxies, so this SNP was excluded from further study. We have added the corresponding explanatory text to the article. (lines 183-186).
- We investigated which traits or clinical significance are associated with the selected SNPs and added the following text to section 4 (lines 226-232): “For most of the SNPs used as instrumental variables in MR analyses, associations with any traits or their clinical significance were not described. An exception is rs1169288 - missense variant of HNF1A (hepatocyte nuclear factor 1-alpha). Its associations with low density lipoprotein cholesterol levels, total cholesterol level, gamma glutamyl transferase levels were found in GWAS and SNP databases (EMBL-EBI, SNPedia). In turn, gamma-glutamyl transferase is involved in maintaining the physiological concentration of glutathione in cells and protects them from damage caused by oxidative stress.». Finally, the following clarification was added (lines 324-327): «Although the SNPs used in this study as instrumental variables do not explain the mechanism of this association, it is likely related to genetic factors. Clinical studies are advisable to validate these associations and assess their practical significance».
- We have added abbreviation explanations:
- Thyroid hormones (TH) – line 67;
- triiodothyronine (T3) – line 80
- free triiodothyronine (fT3) – line 77;
- transient middle cerebral artery occlusion – line 248;
- SRY-box transcription factor 2 – line 249;
- reactive oxygen species (ROS) – line 255
- We corrected term "exchanger" (line 281).
- We capitalized the institute name (line 11)
Once again, thank you for the time, effort and expertise you dedicated to reviewing our work. Please rest assured that we will work on the necessary amendments and look forward to your continued guidance.

Reviewer 2 Report
Comments and Suggestions for Authors
I found this manuscript very reasonably and very accurately prepared.
However, my major question is: the authors have performed a thorough GWAS analysis of the SNPs associated with the level of free triiodothyronine, while in the conclusion, they speak about actually the free triiodothyronine level itself and its association with the decreased risk of unfavorable ischemic stroke outcomes. As it is known, thyroid hormone signaling is mediated by nuclear receptors, which have the ability to significantly change chromatin structure and gene transcription, thus epigenetic and other factors cannot be ruled out.
What I mean, is that the genetically predicted increase in fT3 levels is not the actual increase in fT3 level in clinical practice. So my suggestion is, to consider correcting the conclusion, that it is rather certain fT3 SNPs are associated with certain stroke outcomes but not the fT3 levels themselves as such were not evaluated in this very study.
Author Response
Dear Reviewer,
We would like to express our sincere gratitude for your thorough review and valuable feedback on our manuscript. Your insightful comments have significantly contributed to enhancing the quality of our work. We are also very grateful for your positive assessment of our work.
We deeply appreciate the time and effort you have invested in evaluating our manuscript and are eager to incorporate your recommendations to improve it further.
Thank you very much for your suggession about role of SNP-releted mechanisms in the association between fT3 levels and stroke outcomes.
In our study, we used genetic variants associated with fT3 level obtained from a GWAS analysis by ThyroidOmics Consortium. This study aimed to find loci associated with thyroid function in cohorts of participants with already measured thyroid hormone parameters. We used these genetic variants in Mendelian randomization as instrumental variables associated with a risk factor, in this case, fT3 level.
According toy your suggesison we have added clarifications about the role of the SNP to the text of our conclusion (lines 324-327): «Although the SNPs used in this study as instrumental variables do not explain the underlying mechanism of this association, it is likely related to genetic factors. Further clinical studies are advisable to validate these associations and assess their practical significance».
Thank you once again for your constructive criticism and support. We greatly appreciate your assistance in refining our work!

Reviewer 3 Report
Comments and Suggestions for Authors
This manuscript presents a Mendelian Randomization study investigating the causal association between free triiodothyronine (fT3) levels and ischemic stroke outcomes. The study aims to provide robust evidence for a potential causal link, which could support the use of thyroid hormone analogs to improve stroke outcomes.
The introduction offers only a cursory overview of the principles of reperfusion therapy in cerebral stroke. Unfortunately, these simplifications do not reflect the current understanding and consensus on reperfusion treatment strategies. A more nuanced and accurate description should be included to better contextualize the study.
A major limitation of the study is the relaxation of the genome-wide significance threshold for SNP selection from the standard p < 5×10⁻⁸ to p < 10⁻⁷. Although this decision is explained as necessary due to a limited number of available SNPs, it significantly weakens the instrumental variable assumption, potentially compromising the validity of the MR analysis. This methodological limitation should be more thoroughly discussed. I recommend adding a dedicated section addressing the study's limitations.
The manuscript should clearly outline the nature of the authors' collaboration with the Thyroid Omics Consortium.
More detailed information about the databases used for the MR analysis is necessary, including their composition, access conditions, and any relevant ethical approvals.
The use of data from the Sterenborg et al. dataset also requires clarification. If informed consent from participants was not required, this should be explicitly stated. If consent was obtained, the manuscript should confirm that.
Author Response
Dear Reviewer,
We are writing to express our profound gratitude for your thoughtful and constructive review of our manuscript titled “Causal association of free triiodothyronine level with ischemic stroke outcome: a mendelian randomization study”. Your expertise and meticulous evaluation have been invaluable in strengthening the quaulity and clarity of our work.
We wish to emphasize that we found all your comments to be thoroughly justified and methodologically sound. We fully agree with your suggestions and appreciate the care with which you identified areas for improvement.
In accordance with these comments, we have made the following changes to the article:
Comment 1. The introduction offers only a cursory overview of the principles of reperfusion therapy in cerebral stroke. Unfortunately, these simplifications do not reflect the current understanding and consensus on reperfusion treatment strategies. A more nuanced and accurate description should be included to better contextualize the study.
We have expanded the paragraph about reperfusion therapy (lines 41-57) «Reperfusion therapy significantly improves the rate of vascular recanalization and functional outcomes in patients with acute ischemic stroke. However, a substantial number of patients are either unsuitable for recanalization therapy or experience limited recovery. The gold standard for treating patients with acute ischemic stroke is intravenous thrombolysis with tissue plasminogen activator. However, it is effective only within 4.5 hours after the onset of symptoms. Intravenous thrombolysis is less effective for larger arterial occlusions, achieving recanalization in only 20% of such cases. Mechanical thrombectomy, indicated for occlusion of a large artery in the anterior circulation, is effective for 24 hours. But comparative studies, which examined its effect on long-term stroke outcomes, have shown conflicting results, since less than 50% of patients achieve functional independence [4-6]. Reperfusion restores cerebral blood flow to the hypoperfused brain region surrounding the infarcted core, sparing penumbra cells. The penumbra shrinks over time; it is seen in 90-100% of stroke patients in the first 3 hours after stroke onset and in about 33% of patients 18 hours after stroke onset [7]. For reperfusion therapy to be successful, additional neuroprotective measures must delay cell death until recanalization is achieved. Neuroprotective strategies may potentially address some of the limitations of reperfusion therapy and improve its effectiveness».
Your insights have undeniably elevated the scientific quality and readability of the paper. We are confident that the revisions not only address your concerns but also strengthen the manuscript’s contribution to the field of stroke epidemiology and thyroid research.
Comment 2. A major limitation of the study is the relaxation of the genome-wide significance threshold for SNP selection from the standard p < 5×10⁻⁸ to p < 10⁻⁷. Although this decision is explained as necessary due to a limited number of available SNPs, it significantly weakens the instrumental variable assumption, potentially compromising the validity of the MR analysis. This methodological limitation should be more thoroughly discussed. I recommend adding a dedicated section addressing the study's limitations.
We have expanded the paragraph about limitation, in particular, by discussing the issue of extended thresholds (lines 313-320): Slightly relaxed thresholds, that were applied to MR analysis, may have introduced false positives. However, in our MR analysis, the F statistics of the selected SNPs exceeded 10, indicating the robustness of the instrumental variables. Also, we had not enough information from GWAS studies to use MR Steiger directionality test to definitely rule out reverse causality. Still, MR itself allows with a very high degree of probability to exclude reverse causality. The study only included Europeans, so it may not be fully representative of populations of other ethnicities. Also, we could not completely exclude the possibility of bias due to pleiotropy, despite a range of sensitivity analyses».
Comment 3. The manuscript should clearly outline the nature of the authors' collaboration with the Thyroid Omics Consortium.More detailed information about the databases used for the MR analysis is necessary, including their composition, access conditions, and any relevant ethical approvals.The use of data from the Sterenborg et al. dataset also requires clarification. If informed consent from participants was not required, this should be explicitly stated. If consent was obtained, the manuscript should confirm that.
We have added a paragraph about Data Availability (lines 174-181): «All GWAS summary datasets analyzed during this study are publicly available. Thyroid Omics Consortium provides GWAS summary data for scientific purposes only if the researcher does not attempt to re-identify individual participants included in the summary statistics. Materials from the article of Sterenborg et al. are licensed under a Creative Commons Attribution 4.0 International License. GISCOME datasets are available on The Common Metabolic Diseases Knowledge Portal. All users are welcome to use results from analyses of these data labeled "Open access" to further their research without seeking explicit permission». We also made a clarifications related to datasets we have used (modified Rankin score 0-2 vs. 3-6 → reported as a continuous variable of modified Rankin score at 3 months after stroke (line 25); in the risk of unfavorable (mRs 3-6) stroke outcome → of ordinal mRs scores (line 28).
Once again, we sincerely thank you for your time, expertise, and dedication to improving our work. We are deeply grateful for the opportunity to refine our study under your guidance!

Round 2
Reviewer 3 Report
Comments and Suggestions for Authors
The manuscript has been revised in accordance with my comments, and the authors have adequately addressed the concerns I raised during the evaluation.